# Comparisons of Tidal Currents in the Pearl River Estuary between High-Frequency Radar Data and Model Simulations

**Langfeng Zhu** [1,2,3,†], **Tianyi Lu** [1,2,3,†], **Fan Yang** [4,*], **Bin Liu** [5], **Lunyu Wu** [6] and **Jun Wei** [1,2,3,*]

1   Guangdong Province Key Laboratory for Climate Change and Natural Disaster Studies, School of Atmospheric Sciences, Sun Yat-sen University, Guangzhou 519082, China; zhulf26@mail2.sysu.edu.cn (L.Z.); luty8@mail2.sysu.edu.cn (T.L.)
2   Southern Marine Science and Engineering Guangdong Laboratory (Zhuhai), Zhuhai 519082, China
3   Key Laboratory of Tropical Atmosphere-Ocean System, Ministry of Education, Guangzhou 519082, China
4   Zhuhai Marine Environmental Monitoring Central Station of the State Oceanic Administration, Zhuhai 519000, China
5   Bureau of Hydrology and Water Resources, Pearl River Water Resources Commission of Ministry of Water Resources, Guangzhou 510520, China; lb@pearlwater.gov.can
6   National Marine Environmental Forecasting Center, Beijing 100081, China; wuly@nmefc.cn
*   Correspondence: yangf@scs.mnr.gov.cn (F.Y.); weijun5@mail.sysu.edu.cn (J.W.)
†   These authors contributed equally to this work.

**Abstract:** High-frequency (HF) radar data, derived from a pair of newly developed radar stations in the Pearl River Estuary (PRE) of China, were validated through comparison with in situ surface buoys, ADCP measurements, and model simulations in this study. Since no in situ observations are available in the radar observing domain, a regional high-resolution ocean model covering the entire PRE and its adjacent seas was first established and validated with in situ measurements, and then the HF radar data quality was examined against the model simulations. The results show that mean flows and tidal ellipses derived from the in situ buoys and ADCP were in very good agreement with the model. The model–radar data comparison indicated that the radar obtained the best data quality within the central overlapping area between the two radar stations, with the errors increasing toward the coast and the open ocean. Near the coast, the radar data quality was affected by coastlines and islands that prevent HF radar from delivering high-quality information for determining surface currents. This is one of the major drawbacks of the HF radar technique. Toward the open ocean, where the wind is the only dominant forcing on the tidal currents, we found that the poor data quality was most likely contaminated by data inversion algorithms from the Shangchuan radar station. A hybrid machine-learning-based inversion algorithm including traditional electromagnetic analysis and physical oceanography factors is needed to develop and improve radar data quality. A new radar observing network with about 10 radar stations is developing in the PRE and its adjacent shelf, this work assesses the data quality of the existing radars and identifies the error sources, serving as the first step toward the full deployment of the entire radar network.

**Keywords:** Pearl River Estuary; tidal currents; HF radar; FVCOM; ADCP

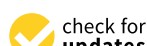



## 1. Introduction

The Pearl River Estuary (PRE), located in the south of Guangdong Province, China, is a trumpet-shaped estuary connecting the delta river network in the north and the South China Sea (SCS) in the south (Figure 1). Pearl River is the second largest and the third-longest river in China, with eight major river inlets into the SCS and an annual discharge of $3.5 \times 10^8$ m$^3$ [1]. The river discharges, along with bottom topography, wind, and tides, complicate the circulations in the PRE and its adjacent seas [2]. Considering the impacts of PRE circulations on human production and lives, especially under the conditions of climate change and natural disasters, a better understanding of the flow fields surrounding the PRE is of great significance to disaster warning and mitigation.

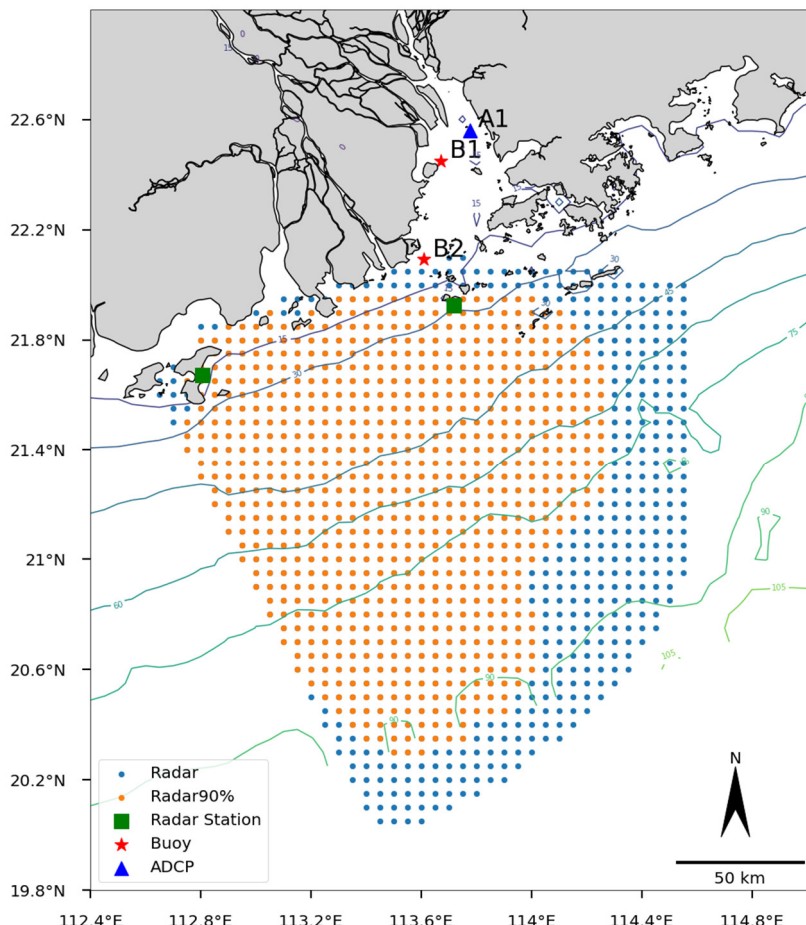

**Figure 1.** Map of the HF radar data domain. Blue spots indicate the original data points and orange spots indicate the data points with a data acquisition rate greater than 90%. The ADCP mooring station is marked by A1, and the two buoys are marked by B1 and B2. The two HF radar stations are marked by green squares.

Previous studies on the flow field characteristics of the PRE and its adjacent waters were mostly based on model simulations [3–6]. Buoys, cruise observations, and tidal gauges were adopted for comparison with the model. The modeling skill assessment suggested that the model can satisfactorily simulate the tidal currents, water level, and salinity structure [2,7]. Outside the PRE, tides propagate into the SCS through the Luzon Strait, and the lunar semidiurnal (M2; 12.42 h period) tidal constituent dominates the currents, followed by K1, O1, and S2 [5,8]. Furthermore, the complexity of topography in the PRE may lead to amphidromic systems in the continental shelf areas [8]. Within the PRE, the Pearl River outflow presents a plume-like structure and forms a corresponding salinity frontal zone which is crucial to substance transport [6,9]. The vertical distribution of the salinity has a seasonal variability (intensified during the summer and relatively weakened during the spring tide in winter) and is affected by the river discharge, as well as wind forcing [4,7,10,11].

High-frequency (HF) radar, a relatively new ocean observing tool, has been used to detect ocean surface currents and waves for decades [12,13]. Bragg scattering, caused by the interaction of electromagnetic waves (wavelength = tens of meters) with the sea surface, was discovered in 1955 [14], which makes it possible for the HF radar to detect the state of the sea beyond the visual range. HF radars are operated in the 3–30 MHz frequency band, ensuring good penetration of the electromagnetic waves through conducting layer. The mechanism of the first- and second-order scattering from the sea surface has been quantitatively explained, providing a solid theoretical basis for the HF radar to detect ocean

surface states [15,16]. HF radars, like other communication systems, adopted the vertical polarization mode which aims to avoid the signal loss caused by the influence of the Earth's impedance. Compared to traditional in situ surface buoys and acoustic Doppler current profilers (ADCPs), HF radars can provide a planar view of current states. They have been widely used in marine environmental monitoring due to their unique advantages such as wide coverage, strong persistence, and all-weather operation [12,17].

The HF radar system can currently perform relatively reliable inversion of ocean currents due to the accurate identification of first-order peaks [15,18]. Many previous studies compared different types of HF radars with mooring observations, proving that the data quality can be affected by environmental factors such as surface wave directions and sea states [19–21]. HF radar measurements can also be examined by cross-validation with model simulations and moorings (e.g., in the New York Bight and Block Island Sound [22–24]).

Usually, a pair of radar stations cover a 200–300 km$^2$ area, therefore a radar observing network with 6–10 radar stations can cover the entire PRE and its adjacent shelf. Our team is building another six radar stations within the PRE. Given that most in situ measurements are mainly deployed within the PRE, the HF radar data on the continental shelf cannot be validated directly against in situ data. In this study, a regional high-resolution ocean model covering the PRE and its adjacent seas was first established and validated with the in situ measurements in the PRE, and the HF radar-derived tidal currents were then examined through comparison with the model results. Furthermore, a set of sensitivity experiments were conducted to explain the role of physical factors in affecting the tidal currents. This study serves as a first step toward the full deployment of the entire radar network, aiming to assess the currently available radar measurements, the model's development, and in situ observations. We note that both radar and model are not perfect, but this work will provide us and readers with very useful information to proceed with the construction and improvement of the entire PRE radar network.

## 2. Materials and Methods

### 2.1. High-Frequency Radar

The array of HF radars has been applied in our study, with an operating frequency of 9.305–9.355 MHz. The number of transmitting antennas at each station is ≥3, and the number of receiving antennas is ≥8 to form an antenna array. This array is capable of detecting the flow field over a large area of the sea in real time. Moreover, it is feasible to build a synchronous network between the different radar stations. The two HF radars adopted in this study were deployed at Wanshan Island and Shangchuan Island (Figure 1), both of which were OSMAR-071G high-frequency surface wave radars operating at a nominal frequency of 9 MHz developed by Wuhan University. The software we used for radar data processing is the SeaMonitor for single-station radial flow. The data processing is divided into a pre-processing module, current extraction module, and radial flow merging module [25,26]. The pre-processing module is mainly to perform channel data validity checks, interference suppression, and channel correction on the basic distance metadata read from the FT1 file, so as to ensure the validity of the processed data and the directional response of the antenna channel to maintain the ideal characteristics. The function of the current extraction module is range-by-distance metadata processing, through spectrum analysis, first-order spectrum separation, azimuth estimation based on the Music algorithm, and other steps, to realize the extraction of radial current velocity. The radial flow merging module aims to combine the results of short-time radial velocity, wind direction, and wave parameters of multiple fields with the median method according to the specified time interval.

The radar coverage ranged from 112.40° to 114.70° E and from 20.20° to 22.00° N, covering an area of approximately 40,000 km$^2$. In the overlapping area observed by the two radars, the longitude and latitude spatial resolution was 0.05° × 0.05°, and the time range was from 12:00 a.m. on 1 March 2020 to 11:50 p.m. on 31 March 2020, with a time resolution of 10 min. Due to the interference of environmental factors, there were missing

values in the synthesized current fields. To ensure the reliability of the data, points with a data acquisition rate of more than 90% were selected (Figure 1).

### 2.2. Mooring Data

The instruments applied to collect the in situ data were one ADCP and two surface buoys. The ADCP was placed at A1 to the northeast of the Pearl River Estuary (113.77° E, 22.56° N), with a time resolution of 10 min and a vertical interval of 1 m between each layer. Due to the existence of blind areas on the surface and the interference of bottom echoes, the ADCP effective measurement depth was 5–11 m. The two buoys were placed at B1 and B2 (113.67° E, 22.44° N, and 113.61° E, 22.09° N) on the west bank of the Pearl River Estuary (Figure 1), with a temporal resolution of 15 min.

### 2.3. Model

The ocean model utilized in this study was the FVCOM model originally developed by the University of Massachusetts Dartmouth (UMASSD) and improved by the efforts of the Woods Hole Oceanographic Institution (WHOI) [27–29]. This model adopts an unstructured triangular mesh to solve the primitive equation through the finite-volume discrete method, which not only retains the high efficiency of finite difference calculation but also incorporates the flexibility of the finite element method to fit the shoreline (http://www.fvcom.smast.umassd.edu/, accessed on 8 March 2022). In addition, time integration of either a mode-split solver or a semi-implicit solver is provided to be chosen. The vertical eddy viscosity and thermal diffusion coefficients are calculated using the General Turbulence Model (GOTM) in the vertical and the Smagorinsky turbulent parameterization in the horizontal [30].

The model grid established in this study, the same as that in [31,32], covered the Pearl River Delta network, the Pearl River Estuary, and the northern South China Sea within a 100 m isobath range, with a minimum grid size of 10 m within the PRE and 2 km resolution in the radar observing area over the continental shelf (Figure 2). The topography data are extracted by Google Earth. The bathymetry data used in the model is acquired from Nautical Chart and bed sweeping data provided by CCCC (China Communications Construction Company) first harbor engineering company limited. In the vertical dimension, a terrain tracking coordinate system combining $\sigma$ and S coordinates was divided into 20 layers, with resolutions up to 0.25 m. The model bottom friction stress is controlled by $\left(t_{bx}, t_{by}\right) = C_d \sqrt{u^2 + v^2}(u, v)$, where $t_{bx}, t_{by}$ are the x and y components of bottom stresses. The drag coefficient $C_d$ is determined by matching a logarithmic bottom layer to the model at a height $z_{ab}$ above the bottom, i.e.,

$$C_d = max\left( k^2 / \ln\left(\frac{z_{ab}}{z_o}\right)^2, 0.0025 \right) \tag{1}$$

where $k = 0.4$ is the von Karman constant and $z_o$ is the bottom roughness parameter. The length scale of $z_o$ set in our model is 0.001.

The river discharges of seven rivers, namely, Xijiang, Tanjiang, Beijiang, Liuxi, Zengjiang, Dongjiang, and Hanjiang, were considered in the model. The climatic river discharge data in the model were based on the daily discharges at the Gaoyao, Shijiao, and Boluo hydrological stations during the 2003–2007 dry season (November to April) provided by the Guangdong Hydrological Bureau. The TPXO8 tidal model, developed by Oregon State University, was utilized as the tidal-driven force at the open boundary [33]. The elevations along the open boundary were calculated by the Tide Model Driver (TMD) using the TPXO8 tidal model. Eight major tidal constituents (M2, N2, S2, K2, K1, O1, P1, and Q1) were included in the tidal drive, which was proven to be reasonable for simulating tidal currents around the PRE [2,8]. Hourly wind fields, as the wind drive, were obtained from Climate Forecast System Version 2 (CFSV2) of the National Centers for Environmental Prediction (NCEP) [34]. The initial fields of temperature and salinity were generated by the

climatology from Simple Ocean Data Assimilation (SODA) (https://climatedataguide.ucar.edu/climate-data/soda-simple-ocean-data-assimilation/, accessed on 1 January 2022). The model was run for 2 months to stabilize the river discharges and model states.

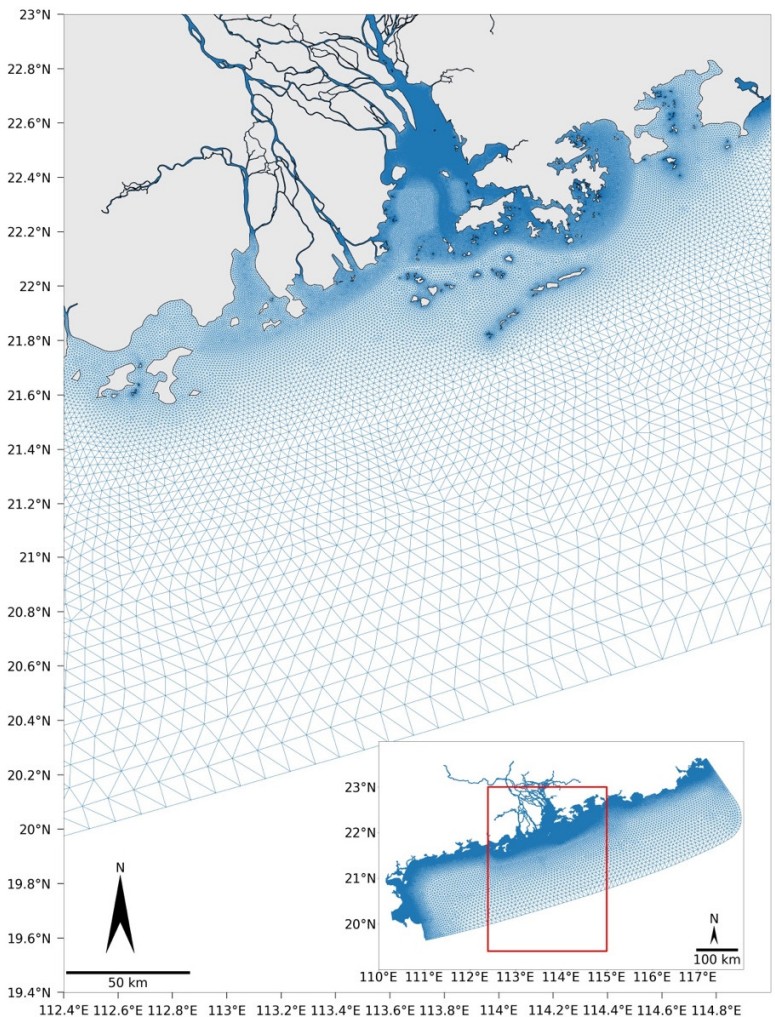

**Figure 2.** The entire model domain (small box), along with a zoomed-in view of the subdomain (red box) in the PRE.

In the observation–model comparisons, the control experiment was performed by applying tidal drive, river discharge, and wind forces. In related sensitivity experiments, one or more external forces listed in Table 1 were removed to assess the effect of several physical factors, including topography, on tidal currents.

**Table 1.** Model setup for the sensitivity experiments.

| Experiment | Wind | Tide | River Discharge |
|:---:|:---:|:---:|:---:|
| CTR | √ | √ | √ |
| No_wind | | √ | √ |
| No_river | √ | √ | |
| No_tide | √ | | √ |
| Tide_only | | √ | |

*2.4. Model Validation*

As in [35], several statistical metrics were included to perform the model–observation comparisons. The mean bias (MB) measures the mean difference between the mooring

observation data and the FVCOM results. The correlation coefficient (CC) estimates the collinearity between their time series. The root mean-squared difference (RMSD) is an absolute measure of the distance between the observational data and the model results. The coefficient of determination (R2) represents the proportion of variance in the response variable explained by the predictor variable. Theskill (WS [36,37]) is a diagnostic index for quantifying the extent to which the model results agree with the observational data, where the WS varies from 0 to 1: WS = 1 indicates perfect consistency between the observational data and the model results, whereas WS = 0 represents no agreement at all. These statistics are defined as follows:

$$MB = \langle m - o \rangle \tag{2}$$

$$CC = \frac{1}{n} \sum_{i=1}^{n} (m_i - \langle m \rangle)(o_i - \langle o \rangle) / (\sigma_m \sigma_0) \tag{3}$$

$$RMSD = \left[ \frac{1}{n} \sum_{i=1}^{n} (m_i - o_i)^2 \right]^{1/2} \tag{4}$$

$$R2 = SSR/SST = 1 - SSE/SST \tag{5}$$

$$WS = 1 - \frac{\langle (m - o)^2 \rangle}{\langle (|m - \langle o \rangle| + |o - \langle o \rangle|)^2 \rangle} \tag{6}$$

where *m* and *o* are the time series (n) for the modeled and observed variables, respectively, and $\sigma_m$ and $\sigma_0$ are their respective standard deviations. Angled brackets $\langle \ \rangle$ denote a mean operator and n is the number of observational samples. SST represents the total sum of squares. SSR represents the regression sum of squares and SSE represents the error sum of squares.

## 3. Results

### 3.1. Comparison between Model Simulations and Mooring Observations

To validate the model performance, the model surface velocity (~1 m) was first compared with the buoys. Figure 3 shows the monthly averaged velocities at stations A1, B1, and B2. We see that the mean flows at A1 and B1 are relatively small, compared to B2. Actually, the tidal currents within the PRE are greater than 1 m/s, with the largest tidal constituent M2 accounting for more than 50% of the total amplitude (see Table 2). By conducting a simple monthly average, the tidal flows are almost filtered out. At B2, the buoy-derived and model mean velocities are consistent, in terms of the amplitude and direction. Station B2 is located right on the main fairway and the exit of the Pearl River, and therefore the mean flow velocity is up to 0.25 m/s.

**Table 2.** Statistics of buoy-derived and model tidal ellipses for $M_2$, $S_2$, $O_1$, and $K_1$ constituents.

| Constituents | Site | Model | | | Buoys | | |
|---|---|---|---|---|---|---|---|
| | | Semi-Major (cm/s) | Semi-Minor (cm/s) | Orientation (°) | Semi-Major (cm/s) | Semi-Minor (cm/s) | Orientation (°) |
| $M_2$ | B1 | 55.83 | 2.66 | 124.21 | 45.60 | 1.21 | 122.24 |
| | B2 | 37.45 | 7.10 | 85.64 | 35.47 | 5.84 | 88.86 |
| $S_2$ | B1 | 29.94 | 0.66 | 125.42 | 24.25 | 0.63 | 122.97 |
| | B2 | 19.75 | 5.23 | 87.03 | 17.12 | 3.29 | 91.68 |
| $K_1$ | B1 | 13.16 | 2.38 | 138.41 | 10.89 | 3.74 | 123.88 |
| | B2 | 12.16 | 2.86 | 86.13 | 13.33 | 2.62 | 85.66 |
| $O_1$ | B1 | 17.50 | 0.81 | 128.68 | 12.12 | 0.61 | 120.28 |
| | B2 | 16.41 | 5.18 | 78.13 | 13.02 | 4.10 | 92.54 |

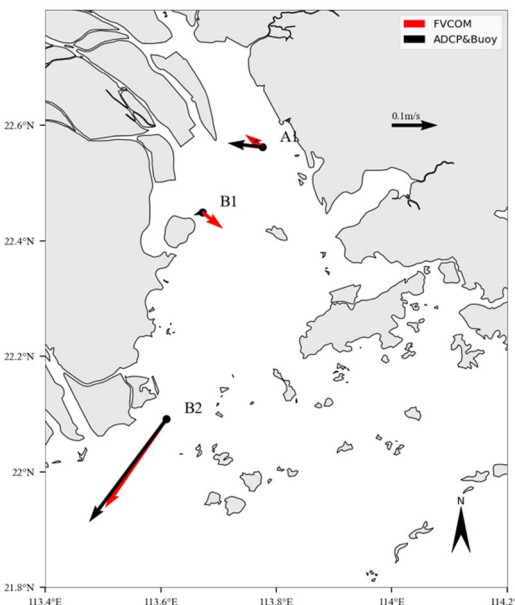

**Figure 3.** Comparisons of monthly averaged velocities (cm/s, vector) from mooring observation and the model.

Figure 4 compares the tidal ellipses of $M_2$, $S_2$, $O_1$, and $K_1$ constituents from the model and buoys. Overall, the model results are in good agreement with the buoy data for all tidal constituents, indicating that the model was able to satisfactorily resolve the tidal currents in the PRE. The largest error appeared at station B1 for the $M_2$ component, with the model semi-major axis (55.83 cm/s) greater than that of the buoy (45.60 cm/s). For $S_2$, $O_1$, and $K_1$ constituents, the averaged errors of the semi-major axis, the semi-minor axis, and the ellipse orientation were 3.2 cm/s, 3.4 cm/s, and 6.9°. The detailed information of the tidal ellipses is listed in Table 2, as well as the statistical metrics in Table 3, showing a very good agreement between the model and the mooring observations.

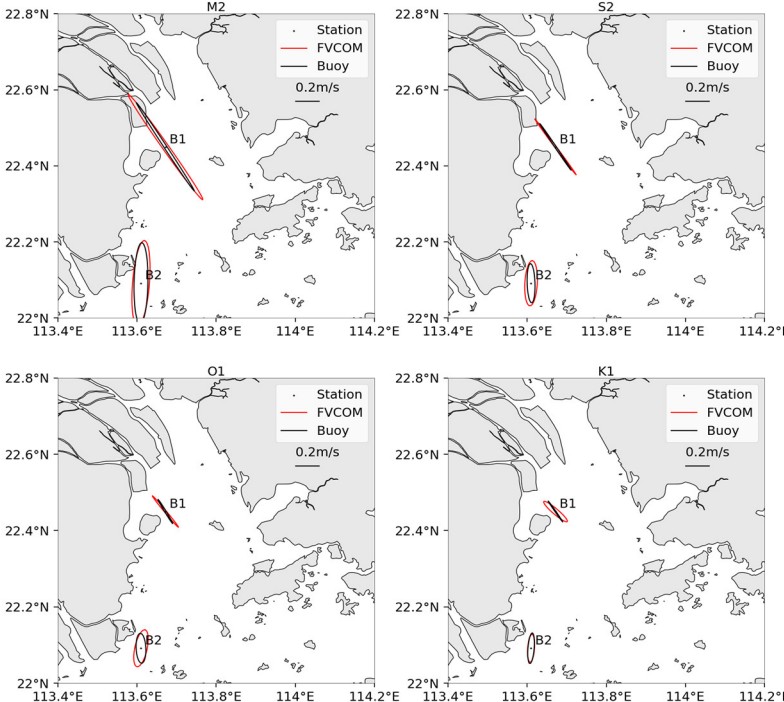

**Figure 4.** Comparisons of buoy-derived and model tidal ellipses for $M_2$, $S_2$, $O_1$, and $K_1$ constituents.

**Table 3.** Statistics of tidal currents (M2, S2, O1, and K1 constituents) for A1, B1, and B2.

| Site | | | Statistical Matrices | | | |
|---|---|---|---|---|---|---|
| | | MB | CC | RMSD | R2 | WS |
| A1 (1st level) | U | 0.0002 | 0.927 | 0.026 | 0.848 | 0.952 |
| | V | −0.0003 | 0.998 | 0.071 | 0.971 | 0.991 |
| B1 | U | 0.0002 | 0.941 | 0.106 | 0.848 | 0.949 |
| | V | −0.0002 | 0.958 | 0.12 | 0.902 | 0.971 |
| B2 | U | −0.0001 | 0.951 | 0.03 | 0.856 | 0.951 |
| | V | −0.0001 | 0.940 | 0.112 | 0.883 | 0.967 |

In this study, the vertical velocity profile was measured only at station A1 by the ADCP. Figure 5 compares the tidal ellipses with depth derived from the ADCP and the model for the $M_2$, $K_1$, $O_1$, and $S_2$ constituents. We can see that the model also satisfactorily simulated the vertical profile of the tidal ellipses for all constituents. It can be noted that the ellipse orientations matched well in the upper layer (e.g., at 5 m depth), but the model ellipses were gradually deflected counterclockwise with depth compared to the ADCP. The largest error was 8.14° for the $M_2$ component at 11 m depth. The orientation errors might be due to the inaccurate topography near station A1, located within the main fairway of the PRE. Moreover, the modeled M2 flow has more tendency of left-land-side rotation toward the bottom, which is from the bottom Ekman effects. This signal is less apparent in the ADCP data, which implies too strong bottom friction for the model at this position. Both surface buoys and ADCP measurements indicated that the $M_2$ tidal constituent is the dominant component of the tidal currents in the PRE. The model was able to reasonably simulate the observed tidal currents and therefore can be used to validate the HF radar tidal currents over the continental shelf, where no in situ observations are available.

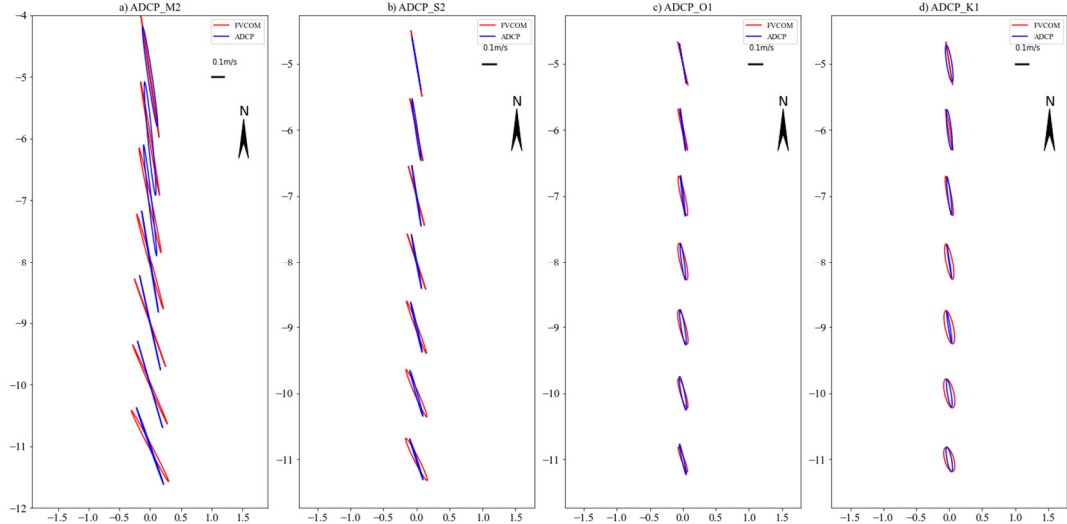

**Figure 5.** Vertical profiles of tidal ellipses ($M_2$, $S_2$, $K_1$, $O_1$) at station A1: model (red) and ADCP (blue).

*3.2. Comparison between Model Simulations and HF Radar Data*

Unlike the in situ pointwise measurements at stations A1, B1, and B2, the HF radar data covered a wide observing area, derived from remotely sensed electromagnetic echoes. Therefore, it is well known that HF radar data may contain uncertain inversion errors due to the environment, noise, inversion algorithms, etc. A comparison between the model and HF radar data can provide a synoptical view of the radar data quality. Figure 6 shows monthly averaged flows from radar measurements and the model. In general, the two flow patterns are similar. A strong along-coast flow appears on the left-hand side of the PRE mouth, with the largest velocity of more than 15 cm/s. This is caused by the Ekman

effects that the northeasterly winds push the flow shoreward, forming the coastal jet. The coastal jet is trapped within ~20 m isobath. Beyond 40 m isobaths, the velocities decrease to ~5 cm/s, flowing toward the coast and merging into the coastal jet. Note that the radar-derived coastal jet does not agree with the model, which will be analyzed in the next section.

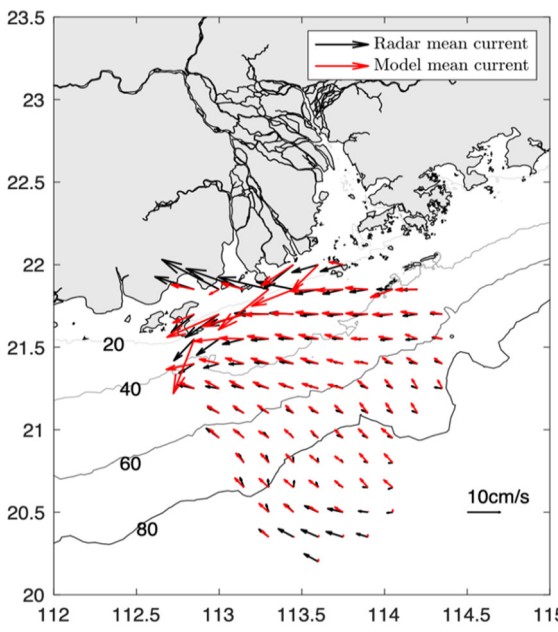

**Figure 6.** Comparisons of monthly averaged velocities (cm/s, vector) from radar measurements and model.

Figure 7a compares the $M_2$ ellipses calculated from the model and the HF radar data. In the model results, the $M_2$ ellipses were overall oriented northwestward. Along the coast to the west shore of the PRE, the ellipses were deformed on different scales, affecting the coastlines and islands. In contrast, the ellipses derived from the HF radar were quite different from the model, indicating that the reflection from the complex coastline prevents HF radar from delivering high-quality information for determining surface currents. Moving seaward, the HF radar and model ellipses were quite consistent between latitudes 21° and 21.5° N, where the averaged water depth was 30–50 m with no islands embedded; thus, the bathymetry effect became insignificant. Below latitude 21 °N, the HF radar ellipses gradually turned northeastward and mismatched the model ellipses.

To explain the discrepancy between the HF radar and model ellipses, we defined an ellipse error as $\sqrt{major\_axis^2 + minor\_axis^2}$ (Figure 7b). The HF radar observing domain could be divided into three areas. In area A, the error was overall significant along the coast, with the largest error of 0.06 m/s. In area B, the HF radar ellipses showed excellent agreement with the model, except for one red spot that was obviously affected by a chain of islands. In area C, the errors increased to 0.04–0.05 m/s, due to the orientation mismatch shown in Figure 7a.

To further diagnose the source of errors, the radar, and model velocities were projected radially according to the two radar stations at Shangchuan and Wanshan Islands (Figure 8). The results show that both Shangchuan and Wanshan stations contained large errors in area A, indicating that coastal islands and bathymetry significantly affected the data quality in both stations. In area B, both stations had relatively small errors, which is reasonable as the overall ellipse errors were small. It is noteworthy that the Shangchuan station showed large errors in area C, whereas the Wanshan station did not. This confirms that the errors in area C (Figure 7b) were mainly from the Shangchuan station.

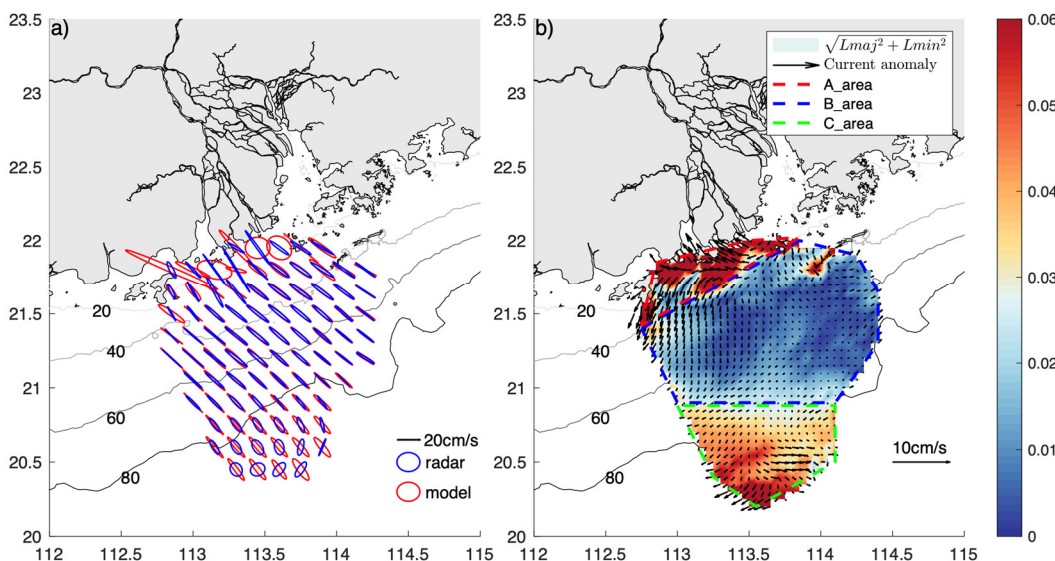

**Figure 7.** (**a**) $M_2$ tidal ellipses for HF radar (blue line) and model (red line) in the radar coverage area. The data are plotted at 0.05° intervals. (**b**) Error distribution of the $M_2$ ellipse amplitude (cm/s, shaded) and current anomaly (cm/s, vector). Isobaths (m) are marked with color-changing solid lines. The current anomaly was derived as $V_{radar} - V_{model}$.

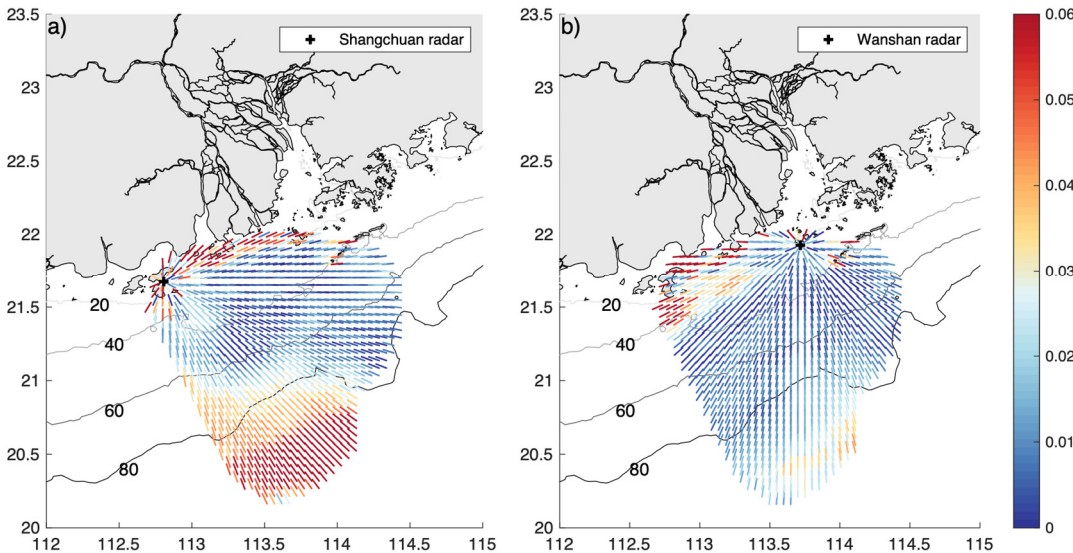

**Figure 8.** Radial error distribution of the $M_2$ ellipse amplitude (cm/s, shaded) attributed to (**a**) the Shangchuan radar station and (**b**) the Wanshan radar station. Isobaths (m) are marked with color-changing solid lines.

Figure 9 shows the scatter plots of the radial velocities derived from the model and the two radar stations. Compared to the model results, the data quality of the Wanshan station was systematically better than that of the Shangchuan station for all three areas. Especially in area C, the radar velocities were underestimated by half compared to the model (Figure 9a). In area A, the tidal currents were dynamically affected by coastlines, topography, and river discharges. We know that the HF radar data were derived from electromagnetic echoes, which were apparently unable to resolve these physical and dynamical factors. In area C, the currents were mainly determined by tides and winds, which can be adequately captured by Doppler spectra; therefore, the HF radar should have been able to resolve the currents in this region. Thus, on the basis of Figure 9a, we infer that the errors of the Shangchuan station were likely caused by inversion algorithms.

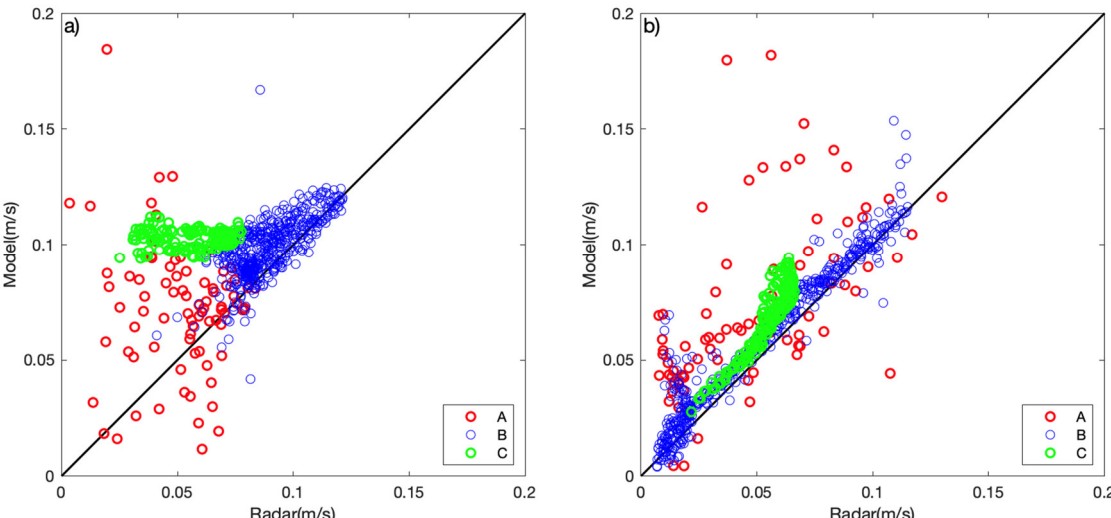

**Figure 9.** Comparison between model and radar radial components of the tidal ellipse at (**a**) Shangchuan and (**b**) Wanshan stations. Points in area A, red circles; points in area B, blue circles; points in area C, green circles.

### 3.3. Sensitivity Model Experiments

To further analyze the characteristics of the tidal currents, we conducted a set of sensitivity experiments to assess the effects of dynamical factors, such as winds, tides, and river discharges. Figure 10 shows the monthly averaged flow fields for the sensitivity experiments. In the control experiment (CTR), there was a strong alongshore current flowing southwestward, driven by all forcings: river discharges, northeast winds, and tides. Apparently, the northeast wind was the dominant forcing, pushing the currents toward the coast. In the no_wind case, the river runoff was dominant, turning to the west of the PRE and forming a plume flow pattern. Compared to the CTR, the alongshore current in the no_wind case was further offshore. In the no_tide case, the flow pattern was very similar to the CTR experiment, because the tide effect was mostly filtered out by monthly averaging. In the no_river case, the wind effect also dominated, except that the river runoff disappeared within the PRE (Figure 10h). To identify the effects of winds, tides, and river discharges, the differences between the CTR and the three sensitivity experiments are shown in the right panel of Figure 8. Obviously, the wind effect was dominant. It produced northwestward currents in the offshore region, an alongshore current in the nearshore region, and an inflow within the PRE, inhibiting river runoff (Figure 10c). As expected, the tide had an insignificant effect on the monthly mean flow. The river discharge showed an outflow from the PRE, turning toward the west shore. It can be noted that both no_wind and no_river cases indicated the effects of river discharge, but their patterns were indeed different. Figure 10b,i present the flow pattern without and with the nonlinear interactions between the winds and river plume, respectively.

The standard deviations are calculated for each sensitivity experiment and the difference between the control and sensitivity experiments are presented in Figure 11. In the control experiment, the standard deviation is high within the PRE related to the strong tidal flow reversal. To the left-hand side of the PRE mouth, the coastal jet also shows a relatively high standard deviation compared to the shelf. Over the shelf, the standard deviation caused by tides and winds becomes weak and uniform, with an amplitude of about 0.3 cm/s. The differences between the control and sensitivity experiments demonstrate the effect of the corresponding factors. We see that the effect of tides is mainly within the PRE and coastal zones, while the river can affect the PRE and the coastal jet. In contrast, the wind effect is dominated on the continental shelf.

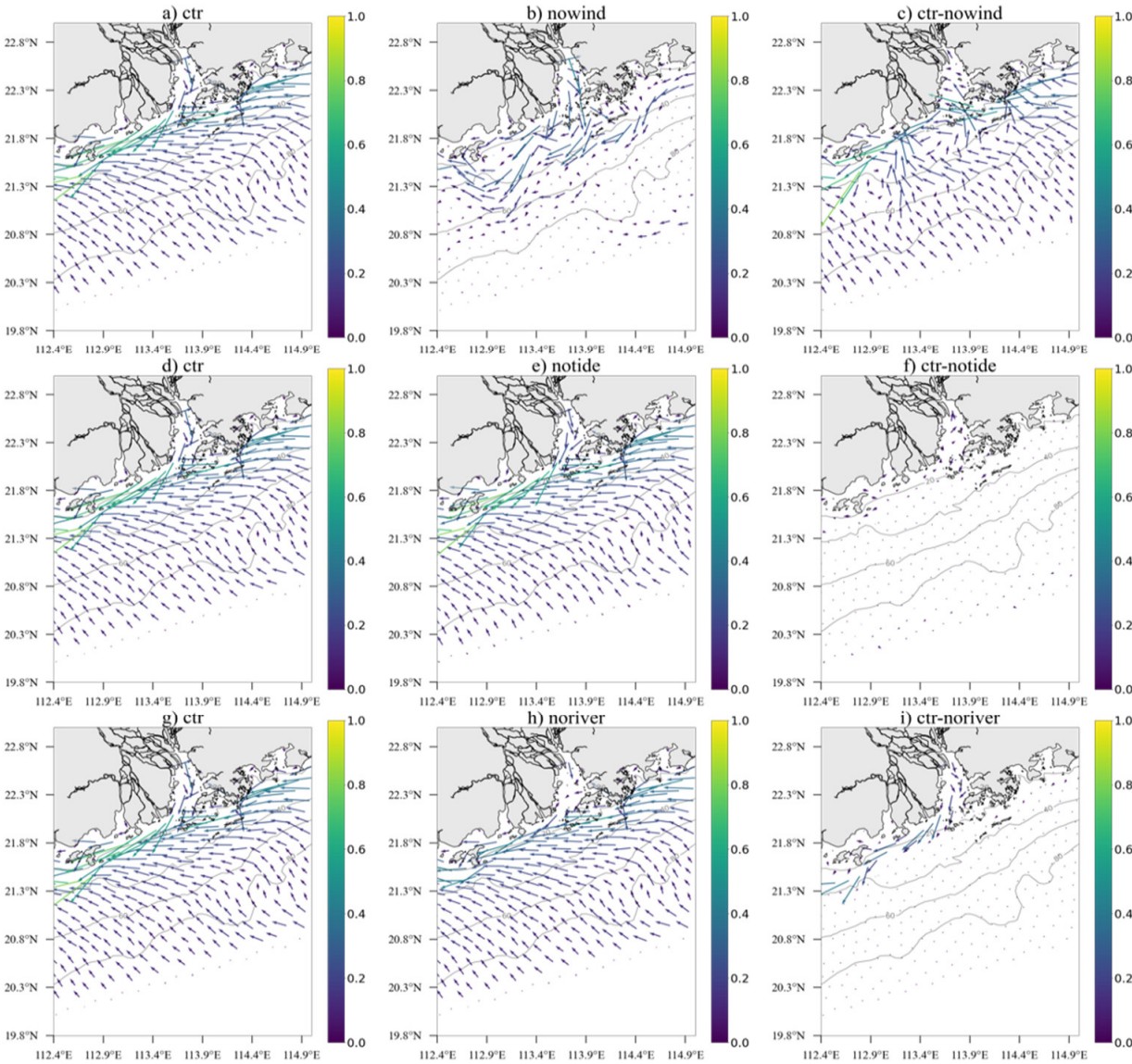

**Figure 10.** The averaged flow in the control experiments (the left column; (**a**,**d**,**g**)), sensitivity experiments (the middle column; (**b**,**e**,**h**), including tests without wind, tide, or river discharge), and the difference between control and sensitivity experiments (the right column; (**c**,**f**,**i**)).

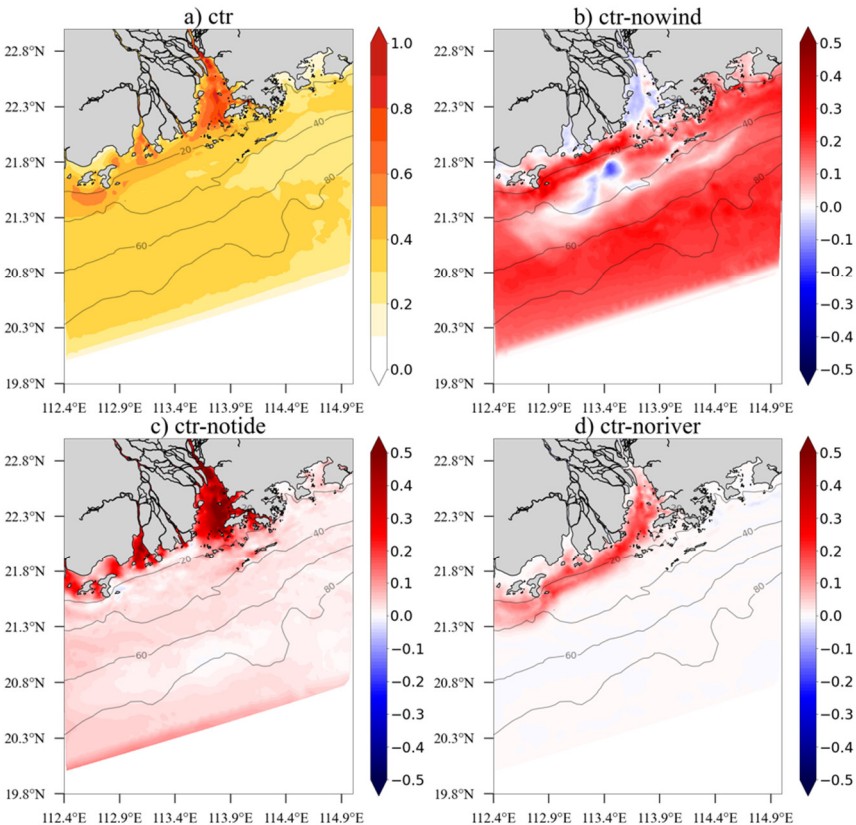

**Figure 11.** The standard deviation (cm/s, shaded) in the sensitivity experiments: (**a**) control experiment; (**b**) the difference between control and no_wind cases; (**c**) the difference between control and no_tide cases; (**d**) the difference between control and no_river cases.

## 4. Conclusions

Since no in situ observations are available in the radar observing domain, a regional high-resolution ocean model covering the entire PRE and its adjacent seas was first established and validated with in situ measurements. The HF radar data quality was then examined against the model simulations. Many previous studies on the flow field characteristics of the PRE and its adjacent waters were based on model simulations [3–6]. In this study, the FVCOM modeling skill assessment suggested that the model can satisfactorily simulate the tidal currents in the PRE, and therefore it is used as a benchmark to assess the HF radar data quality.

With a set of sensitivity experiments, we can diagnose the roles of tides, winds, river discharges, and topography on the radar data quality. Although the radar measurements are derived from the inversion algorithm of electromagnetic echoes, the radar data quality apparently is affected by the above-mentioned physical factors. Based on the sensitivity experiments, the coastal zone to the left-hand side of the PRE mouth is affected primarily by river discharges and tides. These two factors, together with complex coastlines and islands, prevent HF radar from delivering high-quality information for determining surface currents. Over the shelf, the effects of river discharges and tides weakened, and the winds become the major physical factors. The HF radars obtain very high-quality data in area B, but poor data in area C. This does not make sense, as the physical processes in the two areas are almost the same. The radial projection of model–radar comparisons indicates that the main errors in area C are very likely from the Shangchuan radar. We note that the model results are not perfect, but it provides useful information to identify the radar error sources, which can be done by pointwise in situ observations.

In summary, the HF radar provided wide-range measurements of ocean surface currents and waves, while the radar data quality is subject to many potential errors [38],



for example, the environmental noises, the inversion algorithms, the ocean physical factors, local topography, and so on. Given that the radar data were derived from airborne electromagnetic echoes, the local bathymetry, and island effects apparently could not be resolved. Therefore, a machine-learning-based inversion algorithm including traditional electromagnetic analysis and a variety of physical processes is needed to develop and improve radar data quality. So far, different types of machine-learning methods (clustering neural network, convolution neural network, etc.) have been included in the extraction of physical variables from the electromagnetic spectrum [39–43]. In this regard, this study is a first step toward the full deployment of the PRE radar network. It provides a physical analysis of tides, river discharges, winds, and topography, assesses the data quality of the existing radars, and identifies the error sources, all of which is useful information to proceed with the improvement of the radar network.

**Author Contributions:** Conceptualization, J.W.; Data curation, F.Y.; Formal analysis, L.Z. and T.L.; Funding acquisition, J.W.; Investigation, L.Z. and T.L.; Methodology, L.W.; Resources, F.Y.; Supervision, B.L.; Validation, L.W.; Visualization, L.Z. and T.L.; Writing—original draft, L.Z.; Writing—review & editing, J.W. All authors have read and agreed to the published version of the manuscript.

**Funding:** This research was funded by the Key Research and Development Program of Guangdong Province (2020B1111020003), the National Natural Science Foundation of China (41976007, 91958101) and the project supported by Southern Marine Science and Engineering Guangdong Laboratory (Zhuhai) No. SML2020SP009.

**Institutional Review Board Statement:** Not applicable.

**Informed Consent Statement:** Informed consent was obtained from all subjects involved in the study.

**Data Availability Statement:** The data presented in this study are available on request from the corresponding author. The data are not publicly available due to the fact that parts of radar data on oceanic state information are classified.

**Acknowledgments:** We thank the three anonymous reviewers for their valuable suggestions that substantially improved the paper.

**Conflicts of Interest:** The authors declare no conflict of interest.

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
