# Peer review of "Comparisons of Tidal Currents in the Pearl River Estuary between High-Frequency Radar Data and Model Simulations"

_applsci, doi:10.3390/app12136509_

Round 1

Reviewer 1 Report

The manuscript titled "Comparisons of Tidal Currents in the Pearl River Estuary between High-Frequency Radar Data and Model Simulations" was an interesting topic. After reading some comments and questions are as below:

- What is the main input of this research please write carefully.

- The RF radar is not introduced sufficiently, such as a band, polarization etc.

- All the maps like 1 and 2 need a scale bar, and a north array.

- FVCOM model is not well presented is it the forward model? and what is the main algorithm in their background?

- Which software you use for the radar data processing? and what were the preprocessing steps?

- The result is not well presented and the conclusion does not show the main point of research achievement. 

- Why mix the conclusion and discussion? is it a journal format? if not please merge the result and discussion or make the separate section as a discussion. In conclusion, just present the main achievement of the research results. 

Author Response

Thank you for your review. Please see the attached Word file.

Reviewer 2 Report

The title of the paper almost correctly describe the paper content, except that area of analysis - the Pearl River Estuary - is represented by only 2-3 points in some parts of the submitted paper. The PRE itself is not covered by HF radar observations. Main results are presented for the South China Sea in area adjacent to the PRE.

Important question and request to authors is: why the mean (non-tidal) ocean circulation in the research area is excluded from analysis and is not mentioned in paper? Please add the explanation on why only tidal currents are analyzed?

Abstract and conclusion. On lines 18-20 it is stated that "HF radar data were validated through comparison with 1) in situ buoys 2) ADCP measurements and 3)model simulation". Real situation is explained by authors in next lines, that HF radar data are compared with the model simulation results only.  Authors demonstrate some ocean model validation inside of the PRE, using two buoys and one ADCP and based on these three points it is considered that model is "perfect" for tidal currents estimations for all simulation domain.

Lines 23-24. ..."primary physical factors determining the PRE tidal currents are river discharges, winds, and local topography". It seems, that more correctly tides are defined by tidal gravity forces in global ocean, propagates to the coastal waters  and could be only modified by the mentioned factors. Similar sentences could  be found in many places through paper (for example in Line 30).

Line 55 - change "or" to "and"

Line 34 - why tidal elevation was applied only along "lower open boundary"? What are boundary conditions on the right open boundary or it is closed?

Line 192: "HF radar was unable to resolve the coastal bathymetry and islands". By its definition HF radar can not be used for these tasks. Maybe shades of coastal islands, reflection from the complex coastline etc.  prevent HF radar from delivering high quality information for determining surface currents?

Line 223: "area C, orange circles" - replace to "green"

Line 225: Correct is Figure 7, not Figure 8.

Line 260: "Figure 8b,i" - are these correct cases?

Discussion and conclusions: please look all remarks for the Abstract.

Author Response

(The authors gave the same response as above.)

Reviewer 3 Report

See attachments. 

Author Response

(The authors gave the same response as above.)

Round 2

Reviewer 1 Report

Thanks to the author for modifying and improving the text, and figures and adding some new materials. I have no more questions just please check the references and might need English grammatical errors.

Good luck!

Author Response

Response to Reviewer 1 Comments

Thanks again for your comments and suggestions.

We have checked the English grammar and revised the references in the manuscript.

Reviewer 2 Report

Authors significantly updated the paper and in its current form I consider it possible to publish given manuscript after correcting some clearly seen inconsistencies.

1. Authors included in text formulae for used data quality criteria. But in Formulae 4 it is missing power of 2 for the summed expression, (mi-oi)^2. And in Formulae 6 must be removed indexes i in (mi-oi) as the summation operator assume such indexes by definition.

2. Table 3 mentions the site S1. Is it really A1?

Author Response

Response to Reviewer 2 Comments

Thanks again for your comments and suggestions.

Point 1:

Authors included in text formulae for used data quality criteria. But in Formulae 4 it is missing power of 2 for the summed expression, (mi-oi)^2. And in Formulae 6 must be removed indexes i in (mi-oi) as the summation operator assume such indexes by definition.

Response 1. We have revised the mistakes you mentioned in the formula part. Thanks for pointing it out.

Point 2:

 Table 3 mentions the site S1. Is it really A1?

Response 2. Sorry for the inconsistent site symbol. Yes, it is really the A1 site as discussed in the article. We have revised the mistake.